# Equivariant Imaging Biomarkers for Robust Unsupervised Segmentation of Histopathology

**Fuyao Chen**[1,4]      FUYAO.CHEN@YALE.EDU

**Yuexi Du**[1]      YUEXI.DU@YALE.EDU

**Tal Zeevi**[1]      TAL.ZEEVI@YALE.EDU

**Nicha C. Dvornek**[1,2]      NICHA.DVORNEK@YALE.EDU

**John A. Onofrey**[1,2,3]      JOHN.ONOFREY@YALE.EDU

[1] *Department of Biomedical Engineering, Yale University, New Haven, CT*

[2] *Department of Radiology & Biomedical Imaging, Yale University, New Haven, CT*

[3] *Department of Urology, Yale University, New Haven, CT*

[4] *Medical Scientist Training Program, Yale University, New Haven, CT*

**Editors:** Accepted for publication at MIDL 2025

## Abstract

Histopathology evaluation of tissue specimens through microscopic examination is essential for accurate disease diagnosis and prognosis. However, traditional manual analysis by specially trained pathologists is time-consuming, labor-intensive, cost-inefficient, and prone to inter-rater variability, potentially affecting diagnostic consistency and accuracy. As digital pathology images continue to proliferate, there is a pressing need for automated analysis to address these challenges. Recent advancements in artificial intelligence-based tools such as machine learning (ML) models, have significantly enhanced the precision and efficiency of analyzing histopathological slides. However, despite their impressive performance, ML models are invariant only to translation, lacking invariance to rotation and reflection. This limitation restricts their ability to generalize effectively, particularly in histopathology, where images intrinsically lack meaningful orientation. In this study, we develop robust, equivariant histopathological biomarkers through a novel symmetric convolutional kernel via unsupervised segmentation. The approach is validated using prostate tissue micro-array (TMA) images from 50 patients in the Gleason 2019 Challenge public dataset. The biomarkers extracted through this approach demonstrate enhanced robustness and generalizability against rotation compared to models using standard convolution kernels, holding promise for enhancing the accuracy, consistency, and robustness of ML models in digital pathology. Ultimately, this work aims to improve diagnostic and prognostic capabilities of histopathology beyond prostate cancer through equivariant imaging. The code is available at https://github.com/fyc423/SRE_Unsupervised_Segm

**Keywords:** Histopathology, Unsupervised Segmentation, Equivariance, Convolutional Neural Network, Prostate cancer

## 1. Introduction

Histopathology evaluation of hematoxylin and eosin (H&E)-stained tissue specimens through microscopic examination is pivotal in the diagnosis and prognosis of diseases. Through evaluation of tissue architecture, cell morphology, and biomarker expression, pathologists

can accurately diagnose conditions, assess disease progression, and inform treatment decisions. However, traditional manual analysis of histopathological specimens by specially trained pathologists is labor-intensive, expensive, subjective, and prone to inter-rater variability (Karimi et al., 2020), which can impact diagnostic consistency and accuracy. As the volume of digital pathology images continues to grow (Mukhopadhyay et al., 2018), there is a critical need for automated analysis. Machine learning (ML), and in particular deep learning based on computer vision techniques, have shown promising performance on histopathological analysis tasks such as classification (Bulten et al., 2022; Silva-Rodríguez et al., 2020; Kather et al., 2019; Nir et al., 2018a; Han et al., 2017), segmentation (Wilm et al., 2022; Nir et al., 2018a), and analysis (Veta et al., 2019; Aubreville et al., 2023).

Despite the promise of ML in digital pathology, several significant limitations remain (Madabhushi and Lee, 2016). In particular, ML models can struggle with the inherent variability and complexity of histopathology images, which lack explicit meaningful orientation and exhibit diverse staining patterns, tissue structures, and cellular arrangements. Many models are also data-intensive, requiring extensively annotated training sets that are time-consuming and costly to produce. Thus, while supervised learning approaches have shown success, they are limited by the quality and quantity of labeled data available and might not generalize well to new, unseen datasets.

In this work, we address these limitations and make the following contributions: (1) apply equivariant feature learning to learn robust histopathologic imaging biomarkers that are resilient to rotations; (2) develop an unsupervised learning pipeline to perform equivariant biomarker-based segmentation in the absence of manual ground-truth annotations; and (3) validate our approach on a prostate tissue micro-array (TMA) dataset.

## 2. Background

### 2.1. Equivariance in Computer Vision

Equivariance under geometric transformation refers to the property of a function or a model where applying a transformation, e.g. rotation or reflection, to an input and then applying the function yields the same result as first applying the function to the input and then applying the transformation to the function's output. The mathematical definition of equivariance is $T(F[f(x,y)]) = F[T(f(x,y))]$, where $f(x,y)$ is an input signal in 2D space, $T$ is a transformation, e.g. rotation or reflection, and $F$ is a function. The ability to capture equivariant features is important for feature extractors in image processing and pattern recognition, especially for image domains that lack explicitly meaningful orientation, e.g. digital histopathologic images. A robust feature extractor should generalize both invariant and intrinsic information from the data, allowing it to handle variations among different inputs and extract high-level features. Such invariance commonly includes rigid transforms like simple two-dimensional translation, rotation, and reflection transforms. Translational and scaling invariance is more common in natural images that typically have explicit orientation, e.g. a horizon line, and certain radiology images that contain explicit anatomical orientations. In contrast, rotational and reflection equivariance, the ability to maintain consistent output regardless of the angle of input data rotation or reflection, garners attention in histopathology image analysis since this data has no explicit orientation.

Deep learning has greatly benefited from the convolutional neural network (CNN). The translation equivariant characteristic of the CNN enables it to capture similar features in arbitrary input positions while maintaining consistent output. However, CNNs are not equivariant to rotation and reflection transforms. Even a slight rotation or reflection of the image can result in a dramatic degradation of performance in biomedical image classification (Du et al., 2025). For example, a CNN can correctly classify a H&E image as a "tumor", but when rotated by 90 degrees (or even reflected left-right), it incorrectly classifies the same image as "benign". A common practice to address such a problem is to use geometric data augmentation to rotate and flip images during training, which effectively increases the number of training samples. The extremely large number of degrees of freedom (DoFs) (trainable parameters) in modern CNNs allows these networks to effectively compensate for geometric changes in data by accommodating vast numbers of training samples, but the underlying features learned by the network are inherently different despite the image being of the same object. Therefore, there is a need to learn equivariant imaging features that consistently and robustly represent the same images. The equivariant feature learning approach proposed in this project addresses a significant limitation of CNN methods.

## 2.2. Equivariant Feature Learning Approaches

A variety of approaches have been proposed to achieve equivariance in CNNs. *Orientation-aware neural networks* learn orientation information actively during training from the data and use the learned information to re-align the images to their standardized orientation (Jaderberg et al., 2015) or learn this information by aligning all image gradients to a similar orientation (Hao et al., 2022). The rotation equivariant vector field network (Marcos et al., 2016) uses filters of various orientations to generate output in the form of vector fields. These approaches introduce extra learnable parameters to the model, which can lead to potential over-fitting. Such methods also tend to fail to align the input when the input has no specific orientation, e.g. histopathological images. *Rotation-encoded neural networks* encode pre-defined rotation transformations using circular harmonics (Worrall et al., 2017), steerable filters (Cesa et al., 2022; Weiler and Cesa, 2019; Weiler et al., 2018), group-equivalent operations (Cohen and Welling, 2016), or actively rotate the filters during convolution (Zhou et al., 2017). Similar attempts have been made to rotate the filters to gain a rotation-invariant property (Chidester et al., 2019; Linmans et al., 2018) or rotate the feature map (Follmann and Bottger, 2018) obtained by the rotated convolutional filter to embed the feature in four different orientations. Alternatively, some researchers process inputs in various orientations simultaneously to make networks aware of orientation relationships (Cabrera-Vives et al., 2017; Gupta et al., 2020; Yao and Song, 2022; Zhou et al., 2022). These methods are theoretically equivalent to the methods that rotate the filters. However, these methods bring excess size and computational cost to the network as the number of pre-defined angles increases. Meanwhile, these methods show weak performance for the angles that are not pre-defined. *Rotation-equivariant coordinate systems* ensure rotational equivariance by transforming the input data to a different coordinate system, e.g. cyclic coordinate systems (Mo and Zhao, 2024) or polar or log-polar coordinate systems (Esteves et al., 2017; Kim et al., 2020; Paletta et al., 2022). These methods benefit from the property that translation on the polar coordinate system is equivalent to rotation

in the Cartesian coordinate system. However, the polar mapping will naturally result in the loss of the phase information and the image will also be distorted. *Weight symmetric convolution* methods explicitly encode the convolution kernel weights to have symmetric properties (Yeh et al., 2016) such as horizontal reflection (Dzhezyan and Cecotti, 2019) or rotational symmetry (Dudar and Semenov, 2019; Fuhl and Kasneci, 2021) for equivariance. However, the performance of these methods was limited due to small kernel sizes, e.g. 3×3, that hindered the model's ability to learn expressive features. The equivariant feature learning approach used in this paper leverages this rotationally symmetric kernel design strategies but addresses the aforementioned limitations (Du et al., 2025; Zhang et al., 2025). To our knowledge, equivariant feature learning strategies have not been applied to histopathologic image analysis tasks.

## 3. Materials and Methods

### 3.1. Equivariant CNN

To facilitate unsupervised equivariant feature learning (Sec. 3.2), we utilize an equivariant CNN as a feature extractor. To achieve rotational equivariance, we use symmetric rotation equivariant (SRE) convolution (SRE-Conv) kernels (Du et al., 2025), which are centrally symmetric and efficiently parameterized to minimize redundancy. A proof of SRE-Conv's equivariance is provided in the Appendix (Sec. A1). We construct a fully convolutional CNN (SRENet) by replacing all standard, non-equivariant convolution layers in a ResNet18 (He et al., 2016) with SRE-Conv layers. Specifically, we replace ResNet18's convolution layers with SRE-Conv layers using kernel sizes [9,9,5,5] at each of the network's four main stages, respectively. An equivariant pooling layer followed by a 1×1 convolutional layer with a stride of 1 was incorporated to ensure consistent positional convolution. The final classification layer after feature extraction uses a global adaptive pooling operation to ensure that the classifier maintains equivariance. We pre-train SRENet using a supervised learning task to learn equivariant histopathologic imaging features.

### 3.2. Unsupervised Equivariant Feature Learning

Due to SRENet's equivariant design, we can extract equivariant imaging features from any layer of the network. To extract features from SRENet, an input image is fed into the model and we extract the feature maps $\mathcal{F}_L$ from the $L$-th layer of the network. The feature map $\mathcal{F}_L$ is scaled to (128,128). To avoid edge artifacts at the tissue-background boundary, we identified the non-background pixels by creating a tissue mask using intensity thresholding and morphological operations. This mask was used to filter feature maps for pixels corresponding to tissue. We randomly sample $n$ feature embeddings from the valid mask region of $\mathcal{F}_L$. These $n$ features underwent unsupervised K-means clustering to identify K distinct clusters of features.

### 3.3. Unsupervised Segmentation

To segment a given test image, the image is fed into SRENet and we extract the $L$-th layer feature map $\mathcal{F}_L$. This feature map is scaled to (128,128). The feature embeddings at all

masked pixel locations are then fed into the K-means clustering model. The predicted K-means labels are subsequently re-mapped to their original pixel locations within the image using the positional information provided by the mask and then upscaled to the input image size. This process yields a cluster label image for each input image.

## 4. Experiments and Results

### 4.1. Dataset

We evaluate using 50 H&E-stained prostate cancer tissue micro-array (TMA) histopathology images from the public Gleason 2019 Challenge dataset (Nir et al., 2018b; Karimi et al., 2020). Each image acquired at $40\times$ magnification ($\sim5120\times\sim5120$ pixels) is annotated by at least one expert pathologist, segmenting the image into benign and Gleason Grade 3, 4, and 5 categories. This dataset is partitioned into equal halves for training and testing. For model pre-training, we used the histopathologic colon cancer dataset NCT-CRC (Kather et al., 2019), which contains 100,000 non-overlapping image patches from 9 different tissue classes of H&E-stained tissue slides.

### 4.2. Model Implementation and Baseline Comparisons

We used the conventional ResNet18 (ResNet) (He et al., 2016) as the non-equivariant CNN baseline. We further compared SRENet with the state-of-the-art rotation equivariant baseline E2CNN (E2CNN) (Weiler and Cesa, 2019) using a WideResNet-16 (Zagoruyko and Komodakis, 2016) backbone. For pre-training, all models were trained for 50 epochs on the NCT-CRC training dataset with an image size of (224, 224) and a batch size of 24 using SGD optimization with a cosine annealing scheduler and a learning rate of $2 \times 10^{-2}$ and cross-entropy loss. As is standard practice for equivariant feature learning (Worrall et al., 2017; Weiler and Cesa, 2019; Cohen and Welling, 2016), no geometric data augmentation is applied during training to avoid confounding effects. Detailed classification results of this pre-training task can be found in the Appendix (Sec. A6). For TMA feature extraction, we used an image size of (512, 512), resulting in the feature map size of (64, 64) after extraction at the $L = 4$ layer. The feature maps $\mathcal{F}_L$ were resized to (128,128) and flattened for K-means cluster fitting with K=3. All experiments are done with an NVIDIA A5000 GPU.

### 4.3. Evaluation Metrics

To evaluate the robustness of our unsupervised segmentation approach, we calculate the following metrics: (i) the intra-class correlation coefficient (ICC) measures the reliability or consistency of measurements within the same group (McGraw and Wong, 1996); (ii) Cohen's Kappa (Kappa) measure that quantifies measurement agreement for categorical data while accounting for the agreement occurring by chance (Cohen, 1960); and (iii) Dice similarity coefficient. We employed these metrics to evaluate the consistency of K-means cluster label images across 12 rotated versions at 30-degree increments. For each input image, we compute each metric across all post-rotation images where each pixel is assigned a cluster label, to quantify how consistently each pixel retains its cluster label after rotation. We assess significant differences ($\alpha$=0.05) between models by computing Wilcoxon rank-sum tests comparing across result metrics.

Table 1: **Unsupervised Segmentation Quantitative Evaluation.** Intra- and inter-subject segmentation performance (mean ± standard deviation) using equivariant learning (SRENet and E2CNN) and standard convolution (ResNet) evaluated with ICC, Kappa, and Dice. We highlight the best performance with bold.

| Model | Intra-Subject | | | Inter-Subject | | |
|---|---|---|---|---|---|---|
| | ICC | Kappa | Dice | ICC | Kappa | Dice |
| SRENet | **0.92±0.04** | **0.90±0.02** | **0.90±0.03** | **0.91±0.02** | **0.90±0.02** | **0.91±0.03** |
| E2CNN | 0.86±0.06 | 0.80±0.04 | 0.81±0.04 | 0.89±0.03 | 0.85±0.06 | 0.86±0.06 |
| ResNet | 0.85±0.03 | 0.82±0.05 | 0.82±0.06 | 0.83±0.01 | 0.82±0.05 | 0.83±0.06 |

## 4.4. Intra-Subject Rotation Analysis

For each testing subject, features were initially extracted from the TMA image in its original orientation (0 degrees). From the set of valid masked features, a random subsampling of $n=2000$ feature samples was performed. The original TMA image was subsequently rotated at 30-degree intervals, yielding 12 rotations (0 to 330 degrees). Features were extracted from each rotated image, and clustered using the K-means model fitted on the 0-degree orientation image, resulting in 12 segmentation images per subject. To facilitate metric calculations, segmentations were rotated back to their original orientation, providing 12 post-rotation images for each subject. SRENet exhibited higher intra-subject ICC, Kappa and Dice when compared to both E2CNN and ResNet ($p<0.05$) (Tab. 1), indicating superior label consistency following rotation.

## 4.5. Inter-Subject Rotation Analysis

For inter-subject analysis, features were extracted from TMA images of 25 training subjects at their original orientation (0 degrees). $n=500$ feature samples were randomly selected for each subject, yielding a total of $n=12,500$ features across all training subjects. This aggregated feature embedding was used to train the K-means clustering model. For 25 testing subjects, TMA images were rotated at 30-degree intervals from 0 to 330 degrees, generating 12 rotated images per subject. Features from these rotated images were clustered with the trained K-means model, resulting in 12 cluster-labeled images per subject. These cluster-labeled images were then rotated back to their original orientation for ICC calculation. In the inter-subject analysis, SRENet again exhibited higher ICC, Kappa and Dice performance compared to both E2CNN and ResNet ($p<0.05$) (Tab. 1). Both intra- and inter-subject analyses underscore the superior performance of SRENet in maintaining unsupervised cluster-label consistency against rotations when compared to E2CNN and ResNet.

Comparison of the unsupervised segmentation results to ground-truth is challenging when no mapping exists between the pathology labels (Gleason Grade categories) and the cluster labels provided by K-means. An alternative way to evaluate the quality of the unsupervised feature embeddings evaluates feature embedding quality by creating an embedding space from a subset of features, mapping pathologist labels to it, and training a classifier

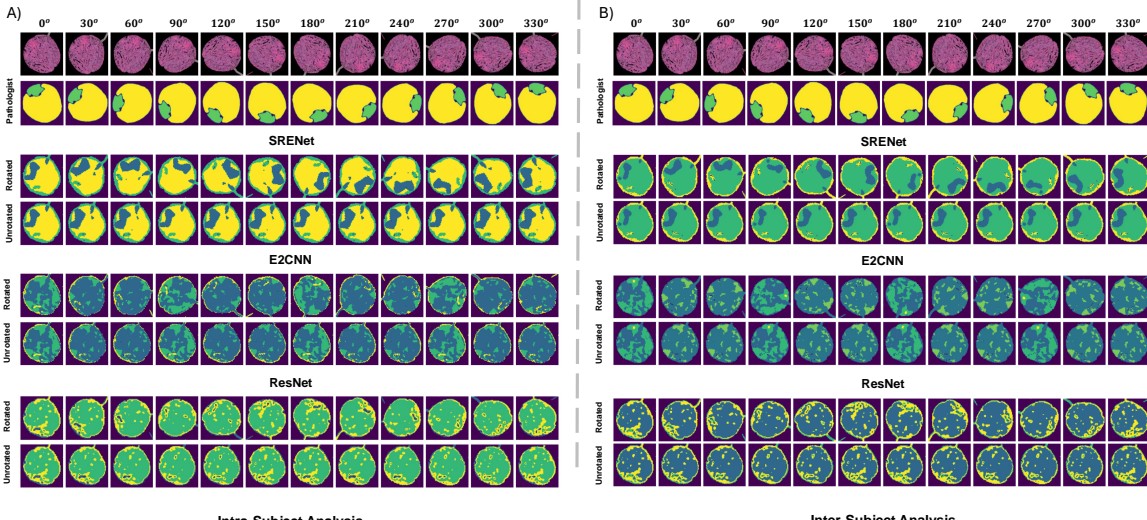

Figure 1: **Intra-Subject and Inter-Subject Analysis.** We visualize an example image for intra-subject (A) and inter-subject (B) analyses using equivariant learning (SRENet and E2CNN) and standard convolution (ResNet). The TMA image undergoes 30-degree rotation increments (top row). For each rotation angle, the resulting segmentation after unsupervised K-means clustering was plotted and then unrotated back to the original input orientation for each model. Because the K-means cluster fitting was performed independently, the segmentation label colormap is not consistent across models.

within this space. The approach then projects all image pixel features to this space and uses the classifier to segment images, effectively assessing how well the unsupervised embeddings align with pathological ground truth. Details of this procedure can be found in the Appendix (Sec. A4). We evaluated the performance of this mapping using Dice similarity coefficient. SRENet demonstrated higher mean±SD Dice values (0.91±0.07) than either E2CNN (0.82±0.12) or ResNet (0.83±0.12) (Appendix Fig. A2) and show example images in Appendix Fig. A3.

### 4.6. Qualitative Evaluation

For unsupervised feature learning, cluster segmentations from SRENet, E2CNN, and ResNet were visualized for intra-subject and inter-subject (Fig. 1) analyses. Our SRENet produces consistent segmentations across rotations, while conventional CNN exhibits changes that hinder meaningful cluster visualization. Although E2CNN preserves segmentation clusters at certain rotation angles, substantial variations occur between these angles. We further compare our clustering results to pathologist segmentation and demonstrate the promising correspondence between pathologist labeling and our unsupervised cluster segmentation method(Fig. 2), highlighting the potential of our method in histopathology applications.

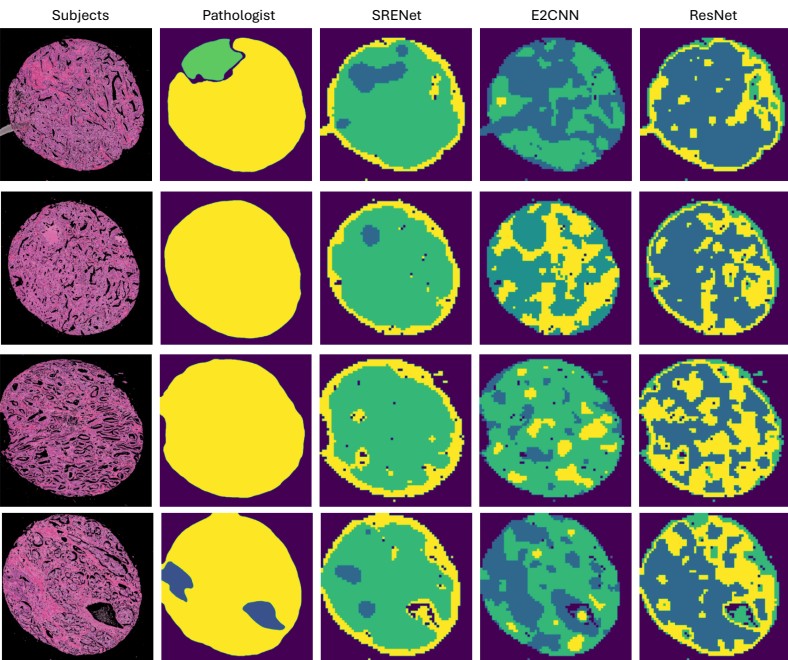

Figure 2: **Comparison to Pathologist Segmentation.** For 4 TMA image example subjects (column 1), we visualize the pathologist segmentations (column 2) in comparison with labeled segmentation maps from our equivariant SRENet model (column 3), rotation equivariant baseline E2CNN (column 4), and conventional non-equivariant baseline ResNet (column 5). Because the K-means cluster fitting was performed independently, the segmentation label colormap is not consistent across models or between models and pathologist segmentations.

To qualitatively assess the equivariant embeddings from pre-training with the NCT-CRC dataset, we visually compare SRENet, E2CNN and ResNet feature spaces using t-distributed Stochastic Neighbor Embedding (t-SNE) (Van der Maaten and Hinton, 2008) in the Appendix (Fig. A5). When images are rotated, the standard ResNet feature embeddings shift significantly within the t-SNE space, mixing labeled clusters, whereas the SRENet feature embeddings stay notably stable, keeping labeled clusters mostly well-separated. E2CNN, the state-of-the-art equivariant method, also exhibits noted shifting in clusters compared to the stable performance of SRENet.

### 4.7. Ablation Studies

We evaluate the performance of K-means clustering with $K = 2, 3, 4$, and Gaussian mixture clustering on the intra- and inter-subject performance using SRENet, E2CNN, and ResNet (see Appendix Sec. A3). Our results (Tab. A2) consistently show that SRENet holds superior performance compared to E2CNN and ResNet regardless of the clustering method employed. Although both intra-subject and inter-subject analyses reveal that as the number of clusters decreases, performance metrics improve, it is likely due to reduced

class complexity and lower chances of incorrect class assignment. While fewer clusters lead to better evaluation performance, they may sacrifice the ability to distinguish between different tissue types. SRENet's robust feature extraction and classification capabilities make it the best performing model in the examined clustering scenarios, and careful consideration is required to balance the number of clusters for optimal application-specific outcomes.

## 5. Discussion and Conclusion

Our study demonstrates that SRENet achieves superior inter- and intra-subject performance for unsupervised segmentation compared to conventional CNN and the state-of-the-art rotation equivariant baseline model. Although completely unsupervised, SRENet shows great potential to align closely with pathologist segmentations (Fig. 2), highlighting the importance of equivariant biomarkers in the analysis of histopathology images intrinsically lacking meaningful orientation. Our method holds promise for identifying unsupervised equivariant biomarkers and has the potential to generalize effectively to other histopathology datasets.

The intra-subject data suggests that our method could be a valuable tool for longitudinal tracking, which is particularly relevant for prostate cancer patients undergoing active surveillance where routine prostate biopsies are collected regularly. Consistent and equivariant biomarkers could be extracted from patient's each biopsy to quantitativel evaluate disease evolution or progression. Furthermore, SRENet could enhance pathological analysis by offering consistent unsupervised segmentation, especially in light of the current variability among raters. This is evident in the TMA dataset used, where the agreement among expert pathologists varies significantly (Cohen's Kappa 0.38 to 0.70) (Karimi et al., 2020). Additionally, SRENet's capability extends to other imaging modalities, indicating its versatility and broad applicability in pathology and beyond. One limitation of this study is the reliance on pre-training using only NCT-CRC images. Domain shift between NCT-CRC colon and TMA prostate datasets may impact model performance.

Other model architectures with potential for rotation equivariance are capsule-based (Sabour et al., 2017) and vision transformer (ViT) (Dosovitskiy et al., 2020) networks. Capsule-based networks achieve rotation equivariance through pose encoding and routing-by-agreement but suffer from high computational demands and optimization challenges due to dynamic routing (Peer et al., 2019; Mitterreiter et al., 2023). As shown in Appendix Table A3, the standard ViT's classification performance on the NCT-CRC dataset drops with rotated test images and performs worse than ResNet without rotation, likely due to the lack of inductive bias and the need for large training datasets, which are often scarce in medical imaging. Replacing the linear projections in ViT with convolutional layers (as in Swin Transformer and Hybrid ViT) for feature extraction also loses rotation equivariance. Furthermore, ViT's positional encoding mechanism disrupts rotation equivariance by encoding positions relative to a fixed frame; even relative positional encoding maintains translational, not rotational, relationships (Chu et al., 2021).

Future work would involve training using prostate-specific datasets or pre-training with a diverse sample of diseases, similar to histopathologic foundation models, and comparing model performance to CNN trained with data augmentation. Moreover, validating SRENet with different feature resolutions at various layer depths of the encoder could further enhance its performance and applicability.

## Acknowledgments

F.C. was supported by the National Institute of Health (NIH) Medical Scientist Training Program Training Grant T32GM007205.

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

## Appendix

## A1. Proof of Circular Kernel Equivariance

In short, the SRE-Conv kernel achieves rotational equivariance with a centrally symmetric kernel, where each circular ring from the center represents one trainable parameter. This design shares values among parameters symmetric to the center, providing local rotational and reflection invariance via the Hadamard product and global equivariance under convolution. Consider a 2D continuous function $f(x, y)$ and central symmetric convolutional kernel $h(x, y)$. Rotation $R(\cdot)$ of their convolution is:

$$R(h * f)(x, y) = R\left(\iint h(u, v) f(x - u, y - v)\, dudv\right) = \iint h(u', v') f(x' - u', y' - v')\, du'dv',$$

where $(x', y') = (x \cos \theta - y \sin \theta, x \sin \theta + y \cos \theta)$ and similarly for $(u', v')$.
Using the linearity of rotation, we establish:

$$\frac{\partial u'}{\partial u} \frac{\partial v'}{\partial v} - \frac{\partial v'}{\partial u} \frac{\partial u'}{\partial v} = \cos^2 \theta + \sin^2 \theta = 1.$$

Thus, we have $du'dv' = dudv$, and

$$R(h * f)(x, y) = \iint R(h(u, v)) R(f(x - u, y - v))\, dudv = R(h) * R(f) = h * R(f),$$

proving rotation equivariance due to $h$'s symmetry. The kernel's translation equivariance ensures the output is equivariant to both global and sub-region rotations and reflections.

## A2. Detailed Quantitative Evaluation and Statistical Testing Results

Figure A1 shows full ICC range for SRENet, E2CNN and ResNet in boxplot. For intra-subject analysis, the median ICC was 0.91 (IQR: 0.89, 0.96) for SRENet. In contrast, E2CNN had a median ICC of 0.86 (0.86, 0.90), while ResNet demonstrated a median ICC of 0.84 (0.82, 0.87). In the inter-subject analysis, the median ICC for SRENet was 0.91, with IQR of 0.90 to 0.92. In comparison, E2CNN showed a median ICC of 0.88 (0.86, 0.91), while ResNet had a median ICC of 0.83 (0.82, 0.84).

## A3. Ablation Study Quantitative Results

Table A2 shows K-means clustering with $K = 2, 3, 4$ and Gaussian mixture clustering on intra- and inter-subject performance using SRENet, E2CNN, and ResNet. SRENet outperforms E2CNN and ResNet across all clustering methods. Performance metrics improve with fewer clusters likely due to reduced class complexity and lower misclassification rates, although too few clusters might limit the ability to distinguish different tissue types. SRENet shows robust feature extraction and classification capabilities make and optimal outcomes require balancing the number of clusters.

Chen Du Zeevi Dvornek Onofrey

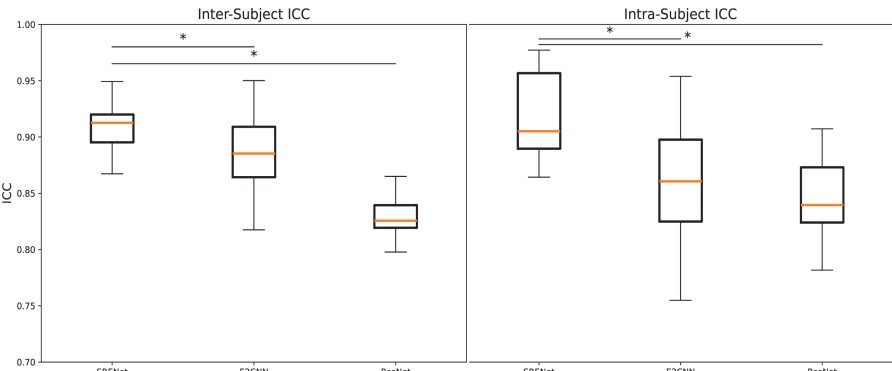

Figure A1: **Intra-subject and Inter-subject ICC Analysis.** Intra- and inter-subject class agreement analyses using equivariant learning (SRENet and E2CNN) and standard convolution (ResNet) evaluated with ICC. * indicates p<0.05.

Table A2: **Ablation Study on Clustering Method.** Intra- and inter-subject segmentation performance (mean ± standard deviation) using variant KMeans clustering ($k = 2, 3, 4$) and Gaussian mixture clustering evaluated with ICC, Kappa, and Dice. We highlight the models reported in the main results with underlines.

| Model | Cluster Method | Intra-Subject | | | Inter-Subject | | |
|---|---|---|---|---|---|---|---|
| | | **ICC** | **Kappa** | **Dice** | **ICC** | **Kappa** | **Dice** |
| **SRENet** | KMeans K=2 | 0.95±0.03 | 0.94±0.02 | 0.95±0.01 | 0.93±0.01 | 0.92±0.02 | 0.94±0.04 |
| | KMeans K=3 | 0.92±0.04 | 0.90±0.02 | 0.90±0.03 | 0.91±0.02 | 0.90±0.02 | 0.91±0.03 |
| | KMeans K=4 | 0.90±0.04 | 0.87±0.03 | 0.87±0.03 | 0.90±0.02 | 0.88±0.02 | 0.88±0.03 |
| | Gaussian Mixture | 0.92±0.04 | 0.90±0.01 | 0.91±0.02 | 0.89±0.02 | 0.89±0.03 | 0.90±0.03 |
| **E2CNN** | KMeans K=2 | 0.90±0.04 | 0.85±0.04 | 0.87±0.04 | 0.92±0.03 | 0.91±0.05 | 0.94±0.05 |
| | KMeans K=3 | 0.86±0.06 | 0.80±0.04 | 0.81±0.04 | 0.89±0.03 | 0.85±0.06 | 0.86±0.06 |
| | KMeans K=4 | 0.81±0.06 | 0.76±0.04 | 0.74±0.05 | 0.80±0.04 | 0.81±0.05 | 0.82±0.06 |
| | Gaussian Mixture | 0.88±0.05 | 0.81±0.04 | 0.82±0.04 | 0.80±0.04 | 0.84±0.05 | 0.86±0.05 |
| **ResNet** | KMeans K=2 | 0.88±0.05 | 0.87±0.04 | 0.89±0.05 | 0.87±0.02 | 0.88±0.04 | 0.89±0.04 |
| | KMeans K=3 | 0.85±0.03 | 0.82±0.05 | 0.82±0.06 | 0.83±0.01 | 0.82±0.05 | 0.83±0.06 |
| | KMeans K=4 | 0.81±0.05 | 0.77±0.05 | 0.75±0.07 | 0.82±0.02 | 0.78±0.05 | 0.77±0.07 |
| | Gaussian Mixture | 0.85±0.05 | 0.83±0.05 | 0.83±0.06 | 0.78±0.03 | 0.82±0.05 | 0.83±0.05 |

## A4. Comparison to Pathologist Segmentation

Comparison of the unsupervised segmentation results to ground-truth is challenging when no mapping exists between the pathology labels (Gleason Grade categories) and the cluster labels provided by K-means. An alternative way to evaluate the quality of the unsupervised feature embeddings involves the following process: (1) define an embedding space (using a principal component analysis utilizing 99% of the cumulative variance) using a small subset of the unsupervised features (100 sample from each subject) from all subjects in the

inter-subject testing cohort; (2) map the ground-truth pathologist labels from this subset onto each point in the embedding space; (3) train a supervised learning classifier (k-nearest neighbor with k=3) within the embedding space; (4) project the feature vectors from all image pixel locations (masked by the ground-truth mask for clarity) to the embedding space; and (5) classify the projected features from each image using the trained classifier to segment the image. The approach effectively evaluates how well the unsupervised feature embeddings map to the ground-truth pathology. We evaluated the performance of this mapping using Dice similarity coefficient. We plot the distribution of Dice values for each method in Fig. A2 and show project pathologist labels onto the example images in Fig. A3.

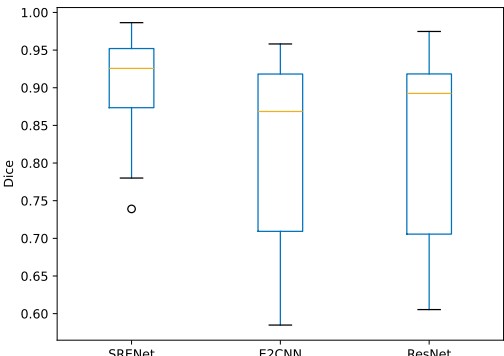

Figure A2: **Quantitative Comparison to Pathologist Segmentation.** We project ground-truth pathologist Gleason Grade labels onto a low-dimensional embedding of model imaging features for each patient in the test set and evaluat using Dice.

## A5. Robust Equivariant Feature Embeddings

We utilized models trained on the NCT-CRC (Kather et al., 2019) dataset to showcase the ability of SRENet to learn stable imaging feature embeddings. We remove the final classification layers of SRENet, E2CNN (Weiler and Cesa, 2019), and ResNet (He et al., 2016), we perform a spatial average pooling on the final feature maps to produce a single vector representations for each image. We then extract feature embeddings from both models for rotated testing set images and applied t-distributed Stochastic Neighbor Embedding (t-SNE) (Van der Maaten and Hinton, 2008) for dimensionality reduction to visually assess these embeddings. The embeddings from SRENet remained stable and well-separated across rotations, unlike those from ResNet, which moved considerably and mixed clusters. SRENet's clusters were also more stable compared to the alternative SoTA equivariant model, E2CNN. These results highlight the robustness of SRENet in maintaining stable feature embeddings crucial for consistent imaging representations.

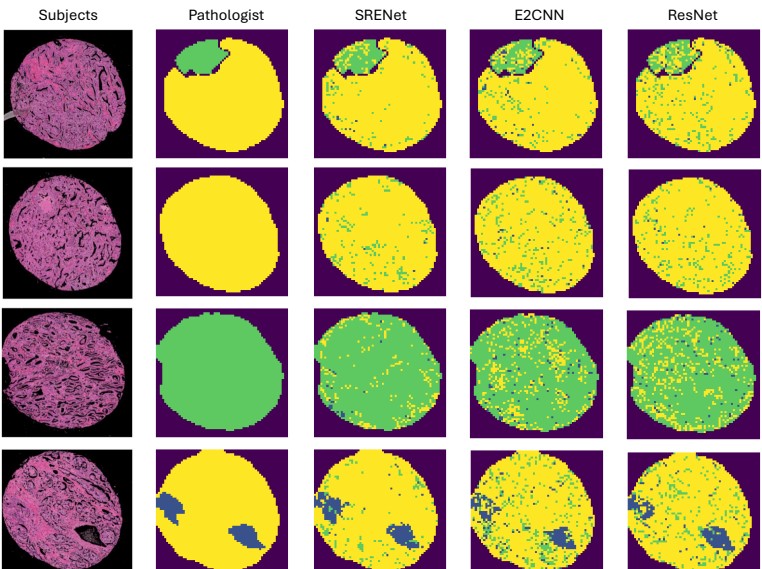

Figure A3: **Qualitative Comparison to Pathologist Segmentation.** We project ground-truth pathologist Gleason Grade labels onto a low-dimensional embedding of model imaging features for each patient shown in Fig. 2.

## A6. Model Pre-training

We pre-train each model on the NCT-CRC (Kather et al., 2019) dataset. NCT-CRC is a colorectal cancer dataset that contains 100,000 training images for 9 different classes and 7,180 test images. We train each model for 50 epochs using the SGD optimizer and cosine annealing scheduler and learning rate of $2 \times 10^{-2}$ with cross-entropy loss. All experiments were done with one NVIDIA A5000 GPU using an image size of $(224, 224)$ with batch size of 24. As is standard practice for equivariant feature learning (Worrall et al., 2017; Weiler and Cesa, 2019; Cohen and Welling, 2016), no geometric data augmentation is applied during training to demonstrate the full capabilities of equivariant learning without introducing confounding effects.

We evaluate model performance by computing classification accuracy on: (1) the original test set without rotation; (2) the rotated test set (rotated by 10° increments; and (3) the reflected test set (horizontal and vertical flips). We report classification results in Table A3. SRENet outperforms E2CNN and ResNet in all test sets. Additionally, we compared CNN model pre-training classification performance to a vision transformer (ViT) (Dosovitskiy et al., 2020) approach. ViT underperforms all other approaches most likely due to its requirements for large amounts of training data. ViT also demonstrates the greatest sensitivity to Rotated data, indicating its limitations compared to equivariant approaches like SRENet and E2CNN. Based on ViT's relatively poor performance on this pre-training task, we excluded ViT as a baseline comparison method for our unsupervised learning segmentation task (Sec. 4.2).

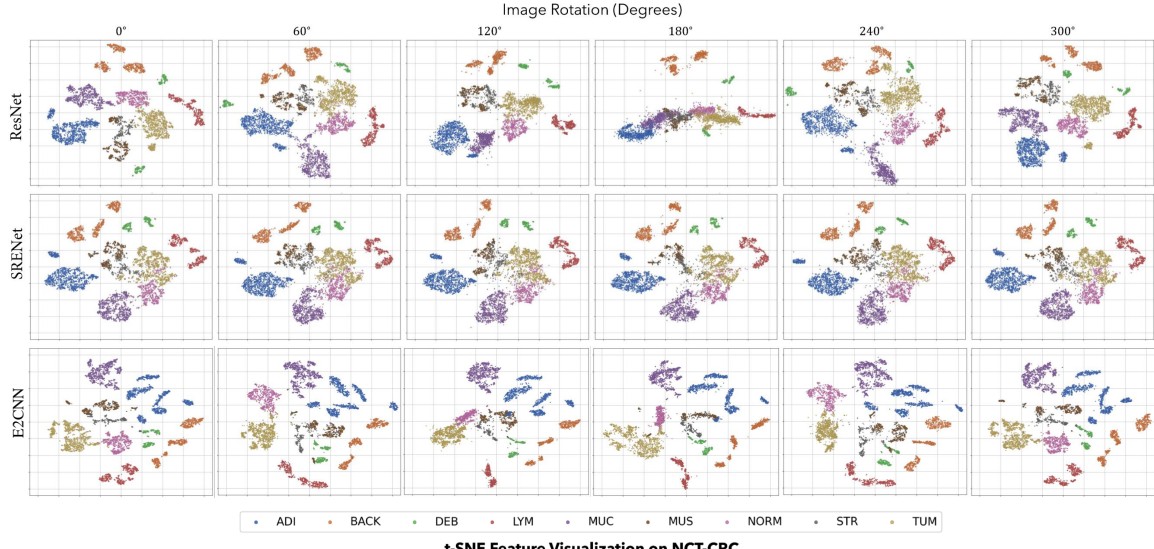

Figure A4: **t-SNE Visualization on NCT-CRC Test Set.** We visualize the clustered test set samples in the NCT-CRC (Kather et al., 2019) dataset using t-SNE. The test input is rotated before feeding into each model. We compare the visualization between (top) standard ResNet (He et al., 2016), (middle) SRENet, and (bottom) E2CNN (Weiler and Cesa, 2019) trained with no rotational augmentation. We colorize the samples in the clustered results according to their classification label.

Table A3: **Pre-training Classification Performance.** Classification accuracy of each model on the original test set, rotated test set, and reflected test set on the NCT-CRC (Kather et al., 2019) dataset. We highlight the best performance with bold.

| Model | Original | Rotated | Reflected |
|---|---|---|---|
| SRENet | **95.5** | **94.8** | **95.5** |
| E2CNN | 93.8 | 92.5 | 93.9 |
| ResNet | 93.7 | 87.3 | 92.9 |
| ViT | 88.4 | 71.8 | 88.5 |

