# OpenReview forum: "Equivariant Imaging Biomarkers for Robust Unsupervised Segmentation of Histopathology"
_MIDL.io/2025/Conference — MIDL 2025 Poster_

### Official Review · Reviewer_MBb9 · 2025-02-20

**Confidence:** 4
**Preliminary Rating:** 4
**Recommendation:** Poster

**Summary:**

The proposed work presents a novel symmetric convolutional kernel for developing robust, rotation-equivariant histopathological biomarkers through unsupervised segmentation. The method is validated on a prostate TMA dataset of 50 patients, demonstrating improved robustness and generalizability compared to SOTA. This work holds potential to enhance the accuracy and consistency of ML models in digital pathology, with applications including prostate cancer, which was investigated in this study.

**Strengths:**

The paper is well-structured and effectively articulates both the existing work and the proposed method. It offers a thorough analysis to support its claims and demonstrates significant potential for advancing ML models in digital pathology. Also, the paper includes a comparative analysis with SOTA methods, providing validation for the proposed approach.

**Weaknesses:**

The paper could benefit from greater clarity in the evaluation section, particularly in terms of explaining the selection of evaluation metrics and providing more detail. Including additional metrics would offer a more comprehensive analysis of the comparisons. Furthermore, the results section would be strengthened by a more thorough explanation of the findings. Providing these additional details would enhance the overall clarity and impact of the paper.

**Detailed Comments:**

- The authors indicate that “A common practice to address such a problem is to use geometric data augmentation to rotate and flip images during training, which effectively increases the number of training samples. The extremely large number of degrees of freedom (DoFs) (trainable parameters) in modern CNNs allows these networks to effectively compensate for geometric changes in data by accommodating vast numbers of training samples, but the underlying features learned by the network are inherently different despite the image being of the same object.” Could the authors provide further clarification on the issue with the new set of features? If the model is appropriately trained, it should be able to consistently extract rotation-invariant features via augmentation. Therefore, what exactly is the issue with the new features?
- The evaluation section could benefit from greater clarity, particularly in explaining the selection of evaluation metrics and providing more detail on their application.
- Including additional evaluation metrics would offer a more comprehensive analysis and a clearer comparison of the proposed method.
- The results section would be strengthened by a more thorough explanation of the findings and their implications. Providing additional details would enhance the clarity and impact of the paper.

**Justification Of The Preliminary Rating:**

The paper is well-written and organized, clearly presenting the problem, methodology, and validation of the proposed work. It provides a thorough analysis and results, addressing key aspects of the research. Overall, it meets the essential requirements of a scientific paper.

**Questions To Address In The Rebuttal:**

Authors are suggest to provide more detail on the evaluation metrics used in this work, offering a deeper discussion of the results, and, if possible, adding additional evaluation metrics for a more comprehensive assessment.

**Special Issue:**

No

---

> ### Author Response · Authors · 2025-03-08
>
> We appreciate your insightful comments and suggestions in enhancing the quality and comprehensiveness of this work. Here are our responses to the specific points raised:
>
> - **Evaluation Metrics:** The Intraclass Correlation Coefficient (ICC) quantifies the similarity between segmented images, emphasizing the reproducibility of the segmentation process. It is especially useful in our context as it assesses the reliability of pixel-wise segmentation consistency across different rotations. To provide a more comprehensive evaluation of segmentation performance, we also incorporate Cohen's Kappa and the Dice coefficient, in addition to ICC. We have provided further introductions on the metrics used in Section 4.3 Evaluation Metrics and showed additional results in Table 1. Specifically, SRENet demonstrated higher performances in all metrics.
> - **Enhance Results and Discussion:** We have included an ablation study in the results section on the clustering method in Appendix Table A2, showing SRENet consistently demonstrating superior performance regardless of clustering techniques employed. We have also expanded our discussion (Section 5) to include other important model architectures in rotation equivariance, including capsule-based network and vision transformer and also included additional details in the potential of SRENet and future work.
> - **Data Augmentation and Feature Extracted:** Data augmentation is commonly used during training to address equivariance issues, but it significantly increases training costs and typically focuses on global rotations, leaving local rotations unaddressed. For example, in histopathology, the local rotation of a cancerous cell or regional structure should not affect its classification or the extracted feature maps, yet global data augmentation fails to address this. Additionally, augmented models learn rotated images as separate examples rather than enforcing rotation-equivariant structures, leading to feature maps that do not consistently transform with rotation. Consequently, the extracted features are not guaranteed to be equivariant (Appendix Figure A4), limiting their effectiveness in providing equivariant unsupervised segmentation for histopathology images via feature learning, as demonstrated by ResNet's performance. Thus, while data augmentation can assist in classification tasks at the expense of training costs, it is limited in providing robust feature learning with respect to equivariance.

---

> ### Comment · Area_Chair_snNY · 2025-03-13
> **Please update your final rating**
>
> Dear reviewer, the MIDL discussion stage will be end by tomorrow March, 14, 2025. Please read the author's rebuttal and update your final rating. Thanks

---

> ### Author Response · Authors · 2025-03-15
>
> Thank you for your thoughtful and encouraging feedback. We truly appreciate your time and effort in reviewing our work. We are thrilled to hear that you found the paper well-written, well-organized, and thorough in its analysis, and we sincerely appreciate the opportunity to address your concerns and improve our contributions based on your insights.
> We are grateful for your recognition of our efforts in clearly presenting the problem, methodology, and validation. Your positive assessment reassures us that our work meets the essential standards of scientific research.
> Once again, thank you for your valuable review and support. We sincerely appreciate it.

---

### Official Review · Reviewer_LMF2 · 2025-02-21

**Confidence:** 4
**Preliminary Rating:** 2
**Final Rating:** 2

**Summary:**

The authors present an unsupervised segmentation of histopathology images using symmetric rotation equivariant imaging biomarkers. In particular, the work introduces SRENet, a convolutional neural network modified with symmetric rotation equivariant (SRE) convolution kernels. The method is designed to extract features that are robust to rotations and reflections from histopathological images. The manuscript demonstrates the method’s efficacy on a prostate tissue micro-array (TMA) dataset.

**Strengths:**

The work addresses a significant limitation of traditional CNNs by incorporating rotational and reflection equivariance.

The motivation of developing SRENet is clearly stated that histopathology images do not have a consistent orientation.

The manuscript presents both intra-subject and inter-subject ICC analyses. The experimental results look promising.

**Weaknesses:**

The novelty of SRENet appears limited, as the authors merely replace standard convolutional layers with SRE-Conv layers without introducing additional innovations. SRE-Conv was previously proposed and is not an original contribution of this work.

The manuscript does not provide sufficient details about SRE-Conv, which is the key technical component of the paper. The authors should elaborate on the design of SRE-Conv and explain why it exhibits symmetric rotation equivariance.

Data augmentation is the most commonly adopted strategy for achieving symmetric rotation equivariance. The authors should compare the performance of a convolutional network trained with data augmentation to that of their proposed method. Additionally, comparing the training times with and without data augmentation could further highlight the advantages of SRENet.

**Detailed Comments:**

The authors need to provide more details on how the SRE-Conv kernels are constructed and parameterized. The authors need to describe how they enforce symmetric rotation equivariant

It would be useful if the authors expand on explaining the choice of the intra-class correlation coefficient (ICC) as the primary metric, and consider including additional segmentation metrics (e.g., Dice coefficient or IoU) to provide a more comprehensive evaluation.

The author need to include a brief discussion or experiment comparing SRENet with a baseline that uses data augmentation, highlighting any differences in performance or training time.

**Justification Of The Final Rating:**

Thank you for addressing some of my earlier concerns with additional evidence. While the application of SRENet to this clinical context is interesting, the manuscript primarily employs established methods with minimal methodological innovation. In its current form, I believe the contribution does not offer sufficient novelty for publication at this conference. As a result, I have maintained my original rating.

**Justification Of The Preliminary Rating:**

The novelty of SRENet appears limited.The manuscript does not provide sufficient details about SRE-Conv. The authors do not compare  SRENet to a critical baseline, i.e., a convolutional network trained with data augmentation.

**Questions To Address In The Rebuttal:**

1. Could the authors provide a more detailed explanation on how SRE-Conv kernels are constructed and constrained to achieve symmetric rotation equivariance? What specific design choices in SRE-Conv distinguish it from existing rotation equivariant approaches?

2. How does SRENet’s performance compare quantitatively to a conventional CNN trained with data augmentation for rotation invariance?
Could the authors include results that isolate the benefits of using SRE-Conv over standard augmentation methods, both in terms of accuracy and training efficiency?

---

> ### Author Response · Authors · 2025-03-08
>
> We appreciate your insightful comments and suggestions for enhancing the quality and comprehensiveness of this work. Here are our responses to the specific points raised:
>
> - **Evaluation Metrics:** The intra-class correlation coefficient ICC assesses the degree of similarity between segments, focusing on the reproducibility of the segmentation process. It is particularly valuable in our context as it measures the reliability of pixel-wise segmentation consistency across different rotations. It evaluates both the consistency (correlation) and conformity (absolute agreement) between images, providing a comprehensive measure of reliability. In addition, we also further included Cohen's Kappa (measures inter-rater agreement for images with categorical labels) and the Dice coefficient. We believe that incorporating these additional metrics enables a more holistic evaluation of our segmentation performance, addressing different aspects of segmentation quality. We have provided further introductions on the metrics used in Section 4.3 Evaluation Metrics and showed additional results in Table 1. Specifically, SRENet demonstrated higher performances in all metrics.
> - **SRE-Conv Kernel:** We achieve rotational equivariance by using a centrally symmetric kernel. In this approach, each circular ring radiating from the center of the kernel acts as a trainable parameter. All parameters that are symmetric around the center share the same value. This configuration ensures local rotational and reflection invariance when applied with the Hadamard product, and global rotational and reflection equivariance under convolution as the kernel moves through the input. For a 2D function with a central symmetric kernel, rotating their convolution results in the same outcome as convolving the kernel with the rotated function. This is due to the kernel's inherent symmetry, which makes it invariant to rotation. Consequently, the kernel's translation equivariance ensures that it maintains equivariance not only for the entire input but also for any rotational or reflective transformations of its sub-regions. More details and equations are provided in Appendix A1 Proof of Circular Kernel Equivariance.
> - **Data augmentation:** Data augmentation is commonly used during training to address equivariance issues, but it significantly increases training costs and typically focuses on global rotations, leaving local rotations unaddressed. For example, in histopathology, the local rotation of a cancerous cell or regional structure should not affect its classification or the extracted feature maps, yet global data augmentation fails to address this. Additionally, augmented models learn rotated images as separate examples rather than enforcing rotation-equivariant structures, leading to feature maps that do not consistently transform with rotation. Consequently, the extracted features are not guaranteed to be equivariant (Appendix Figure A4), limiting their effectiveness in providing equivariant unsupervised segmentation for histopathology images via feature learning, as demonstrated by ResNet's performance. Thus, while data augmentation can assist in classification tasks at the expense of training costs, it could be limited in providing robust feature learning with respect to equivariance. We greatly appreciate the reviewer’s insights and suggestions and will include future experiments to compare feature extraction and unsupervised learning performance of models pre-trained with data augmentation to SRENet for training time, segmentation accuracy, and equivariance robustness to provide a more thorough evaluation for the scientific community.
> - **Novelty of Equivariant Unsupervised Learning:** As aforementioned, we extend our work beyond the classification task by utilizing the SRE-Conv kernel for equivariant feature learning, tackling the challenging task of unsupervised segmentation. This is particularly crucial in the medical imaging domain, where expert labels are often scarce, and inter-rater variability poses significant challenges. This work develops a framework for equivariant feature-based unsupervised segmentation, effectively demonstrating the correspondence between unsupervised feature embedding maps and the ground-truth pathology.

---

> ### Comment · Area_Chair_snNY · 2025-03-13
> **Please update your final rating**
>
> Dear reviewer, the MIDL discussion stage will be end by tomorrow March, 14, 2025. Please read the author's rebuttal and update your final rating. Thanks

---

> ### Author Response · Authors · 2025-03-15
>
> Thank you for your time, thoughtful feedback, and for engaging with our work. We sincerely appreciate the opportunity to address your concerns and improve our contributions based on your insights.
> We would like to highlight that this work aligns with the conference’s focus on advancing deep learning applications in automated image analysis and innovative clinical approaches for disease diagnosis. By moving beyond classification tasks and developing a method that holds promise to reduce reliance on expert annotations while improving segmentation accuracy, our approach has the potential to support the broader vision of integrating AI-driven solutions into clinical workflows.
> Once again, thank you for your careful review and constructive feedback. We truly value your time and effort and are grateful for the opportunity to refine our work.

---

### Official Review · Reviewer_u1YC · 2025-02-24

**Confidence:** 4
**Preliminary Rating:** 3
**Final Rating:** 5

**Summary:**

In histopathology imagery, there is no notion of "up". Therefore, CNNs which are highly sensitive to orientation, fail to generalize well in this unique domain. The authors propose a method of equivariant feature learning to significantly improve segmentation of histopathology imagery. To achieve this, the authors propose to construct an SRENet version of a ResNet-18. The authors perform experiments on 50 H&E-stained prostate cancer tissue micro-array (TMA) histopathology images from the public Gleason 2019 Challenge dataset.

**Strengths:**

1. The paper is well motivated. CNNs struggle significantly with rotation invariance and equivariance. And this is a problem domain where, somewhat uniquely, there is no canonical orientation.

2. The authors methods are sound and demonstrate a significant improvement over baseline CNNs, especially in relation to intra-observer agreement.

**Weaknesses:**

1. While the authors focus on CNNs for obvious reasons, a lot of the computer vision community has moved nearly completely away from CNNs. Their replacements, first capsule networks then transformers, both show significantly better ability to handle the affine transformations on input orientation. For example, in Capsules for Biomedical Image Segmentation, the authors showed a capsule-based segmentation network dramatically outperformed a CNN-based segmentation network at various degrees of rotation on the input of images. Transformers share this property, as both represent features as vectors rather than scalars and affine transformations are handled by the linear projections of the parent capsules or qkv multiplications. It would have been to see if this approach can beat out these more advanced techniques and give CNNs some more footing again in the modern era.

2. The authors use metrics to examine the consistency of predictions. However, it would have been great to measure the quality of segmentations produce (via Dice). I understand the authors say in many cases there is no ground truth segmentation as they're costly to produce, but there are ground-truth segmentations for this dataset.

**Detailed Comments:**

No minor comments. Please see the questions and weaknesses sections.

**Justification Of The Final Rating:**

The authors provided a fantastic and thorough response to my concerns. The additional discussions around transformers and capsule networks as well as the additional experiments around the quality of segmentations were both sorely missing in this paper and I think it significantly elevates the quality. I am raising my rating all the way from Borderline to Strong Accept. In a world of nothing but Transformers, it is great to see a CNN-based approach shine and I think this would be of significant value to the community to re-think where CNNs might still have an important place.

**Justification Of The Preliminary Rating:**

The authors paper is well motivation in that it is fixing a crucial issue with CNNs in a problem domain where this is uniquely required. However, the authors fail to compare with or even acknowledge newer capsule-based or transformer-based approaches which also largely solve the rotation issue. At the very least these should be acknowledged, and preferably compared with.

**Questions To Address In The Rebuttal:**

The authors can still utilize the motivation of CNNs fail to handle rotations, while acknowledging that capsule networks and transformers largely solved this problem already. At least an acknowledgment in the rebuttal might be enough to push to weak accept, but more preferably the authors would compare against these techniques and use this as an opportunity to give CNNs a new life in the modern era, similar to how ConvNeXt did.

---

> ### Author Response · Authors · 2025-03-08
>
> We appreciate your insightful comments and suggestions in enhancing the quality and comprehensiveness of this work. Here are our responses to the specific points raised:
>
> - **In Comparison to Capsule-based network and Vision Transformer:** We appreciate the reviewer’s valuable insights regarding rotation equivariance potential of capsule-based and vision transformer (ViT) networks, and agree it is critical to include these important model architectures in our discussion. For capsule-based networks, while the model design enables rotation equivariance via direct pose encoding and routing-by-agreement mechanism, the dynamic routing process is computationally intensive, leading to increased training times and resource demands and also makes optimization challenging. Regarding ViT, while the original proposed ViT splits the image into fixed-size patches and projects them using a linear transformation, many current implementations replace the linear projection with a convolutional layer for improved feature extraction (e.g. Swin Transformer, Hybrid ViT), which lacks rotation equivariance. Furthermore, the positional encoding mechanism inherently breaks rotation equivariance because it encodes positions relative to a fixed reference frame. While relative positional encoding could improve generalization, it still operates in an axis-aligned manner, preserving translational relationships but not rotational relationships. In Appendix Table A3, we show that vanilla ViT classification performance in NCT-CRC dataset decreases with rotation of testing images (88.4% accuracy on unrotated images down to 71.8% accuracy on rotated images). Furthermore, ViT performance without rotation is lower compared to baseline ResNet (93.7% accuracy on unrotated images down to 87.3% accuracy on rotated images), which likely is due to lack of inductive bias and need of large training data. While natural image datasets might fulfill this requirement, there often lacks large enough medical datasets for proper training or pretraining of ViT. SRENet's performance on this task is much high (95.5% accuracy on unrotated images down and 94.8% accuracy on rotated images). We further included this discussion in Section 5.
> - **In comparison to pathology ground truth:** Comparison of the unsupervised segmentation results to ground-truth is challenging when no mapping exists between the pathology labels (Gleason Grade categories) and the cluster labels provided by K-means. An alternative way to evaluate the quality of the unsupervised feature embeddings involves the following process: 1) we define an embedding space (using a principal component analysis utilizing 99% of the cumulative variance) using a small subset of the unsupervised features (100 sample from each subject) from all subjects in the inter-subject testing cohort; 2) we map the ground-truth pathologist labels from this subset onto each point in the embedding space; 3) train a supervised learning classifier (k-nearest neighbor with k=3) within the embedding space; 4) project the feature vectors from all image pixel locations (masked by the ground-truth mask for clarity) to the embedding space; and 5) classify the projected features from each image using the trained classifier to segment the image. The approach effectively evaluates how well the unsupervised feature embeddings map corresponds to the ground-truth pathology. We evaluated the performance of this mapping using Dice similarity coefficient.  SRENet demonstrated higher meanSD Dice values (0.91$\pm$0.07) than either E2CNN (0.82$\pm$0.12) or ResNet (0.83$\pm$0.12). We have added a description to Sec. 4.5 with details in Appendix Sec. A4. We plot the distribution of Dice values for each method in the Appendix Fig. A2 and show example images in Appendix Fig. A3.

---

> > ### Comment · Reviewer_u1YC · 2025-03-15
> > **Great rebuttal**
> >
> > Fantastic and thorough response. The additional discussions and experiments really raise the quality of this paper and address concerns that I'm sure many readers would have shared (e.g. why should I read a paper about CNNs in 2025?).

---

> > ### Author Response · Authors · 2025-03-15
> >
> > Thank you for your thoughtful and encouraging feedback. Your insights greatly helped us improve our paper. We truly appreciate the opportunity to address the missing discussions and experiments. Your suggestions on transformers, capsule networks, and segmentation quality were invaluable in refining our work.
> > We share your sentiment that CNNs still have an important role in deep learning, and we are thrilled that our work can contribute to this discussion within the community.
> > We sincerely appreciate your time and effort in reviewing our paper. Your constructive feedback made a significant difference, and we are grateful for your endorsement.

---

> ### Comment · Area_Chair_snNY · 2025-03-13
> **Please update your final rating**
>
> Dear reviewer, the MIDL discussion stage will be end by tomorrow March, 14, 2025. Please read the author's rebuttal and update your final rating. Thanks

---

### Official Review · Reviewer_eB6K · 2025-02-27

**Confidence:** 3
**Preliminary Rating:** 3
**Final Rating:** 3

**Summary:**

The authors introduce SRENet, a CNN architecture that replaces traditional convolution layers with symmetric rotation equivariant (SRE) convolution kernels to address rotation problems in histopathology images. Then, the method leverages unsupervised K-means clustering on the feature embeddings for segmentation.

**Strengths:**

Figure 1 and Figure 2 show superior rotational consistency through high ICC values and visual comparisons against both standard CNNs and state-of-the-art rotation-equivariant models.

The article eliminates the need for extensive manual annotation by applying unsupervised K-means clustering for robust feature embedding.

**Weaknesses:**

In Figure 2, the absence of pathologist ground truth makes it challenging to assess the clinical relevance of the segmentation results. Moreover, the use of different colormaps in Figures 2 and 3 complicates the direct comparison of the outputs from the various models.

One notable limitation is the absence of experiments directly comparing the model's segmentation outputs with pathologist-provided segmentations.

**Detailed Comments:**

Include a brief analysis on the sensitivity of K-means clustering. Or more experiments on different clustering methods.

**Justification Of The Final Rating:**

Dear authors, Thanks for your response. Your response addressed part of my concerns. However, the major innovation (SRE convolution) seems from previous accepted paper (ISBI 2025). I will keep my final rating at 3.

**Justification Of The Preliminary Rating:**

It is recommended to add a direct comparison experiment in the experimental section to compare all the model segmentation results with the segmentation results of pathologists.

More analysis on the sensitivity of K-means clustering.

**Questions To Address In The Rebuttal:**

How do SRENet and baseline methods perform in terms of segmentation accuracy (e.g., Dice score, IoU) when compared to pathologist annotations?
How do E2CNN perform in t-SNE Feature Visualization on NCT-CRC?

**Special Issue:**

No

---

> ### Author Response · Authors · 2025-03-08
>
> We appreciate your insightful comments and suggestions in enhancing the quality and comprehensiveness of this work. Here are our responses to the specific points raised:
> - __Sensitivity Analysis of K-means Clustering:__ We have included a detailed analysis of the sensitivity to the number of clusters in K-means and another clustering technique (gaussian mixture) in Section 4.7 (see Appendix Table A2 for quantitative results). Our findings indicate that the superior performance of SRENet was not substantially impacted by the choice of clustering techniques.
> - __E2CNN Performance in t-SNE Feature Visualization on NCT-CRC:__ We have added the performance of E2CNN in t-SNE Feature Visualization on NCT-CRC to Appendix Figure A4. While relatively stable compared to ResNet, in contrast to SRENet, some of the clusters in E2CNN still exhibited movement and mixing.
> - __In comparison to pathology ground truth:__ Comparison of the unsupervised segmentation results to ground-truth is challenging when no mapping exists between the pathology labels (Gleason Grade categories) and the cluster labels provided by K-means. An alternative way to evaluate the quality of the unsupervised feature embeddings involves the following process: 1) we define an embedding space (using a principal component analysis utilizing 99% of the cumulative variance) using a small subset of the unsupervised features (100 sample from each subject) from all subjects in the inter-subject testing cohort; 2) we map the ground-truth pathologist labels from this subset onto each point in the embedding space; 3) train a supervised learning classifier (k-nearest neighbor with k=3) within the embedding space; 4) project the feature vectors from all image pixel locations (masked by the ground-truth mask for clarity) to the embedding space; and 5) classify the projected features from each image using the trained classifier to segment the image. The approach effectively evaluates how well the unsupervised feature embeddings map correspond to the ground-truth pathology. We evaluated the performance of this mapping using Dice similarity coefficient.  SRENet demonstrated higher meanSD Dice values (0.91$\pm$0.07) than either E2CNN (0.82$\pm$0.12) or ResNet (0.83$\pm$0.12). We have added this description to Sec. 4.5 with details in Appendix Sec. A4. We plot the distribution of Dice values for each method in the Appendix Fig. A2 and show example images in Appendix Fig. A3.

---

> ### Comment · Area_Chair_snNY · 2025-03-13
> **Please update your final rating**
>
> Dear reviewer, the MIDL discussion stage will be end by tomorrow March, 14, 2025. Please read the author's rebuttal and update your final rating. Thanks

---

> ### Author Response · Authors · 2025-03-15
>
> Thank you for your time, thoughtful feedback, and for engaging with our work. We truly appreciate the opportunity to address your concerns and refine our contributions based on your insights.
> We hope to highlight that this work aligns with the conference’s focus on advancing deep learning applications in automated image analysis and innovative clinical approaches for disease diagnosis. By extending beyond classification tasks and developing a method with the potential to reduce reliance on expert annotations while enhancing segmentation accuracy, our approach holds promise to contribute to the broader vision of integrating AI-driven solutions into clinical workflows.
> Once again, thank you for your careful review and constructive feedback. Your time and effort are truly appreciated, and we are grateful for the opportunity to improve our work.

---

### Author Rebuttal · Authors · 2025-03-08

**Rebuttal:**

Dear Reviewers and Area Chairs,

Thank you sincerely for your thoughtful and detailed reviews. We greatly appreciate your time and valuable feedback.
We are encouraged by your positive remarks. You highlighted the following strengths:
- Superior rotational consistency demonstrated by our method;
- The potential elimination of extensive manual annotation via unsupervised learning;
- A clear motivation given the orientation challenges in histopathology images;
- Our method's soundness and significant improvements over baseline methods; and
- Our well-structured presentation and thorough analysis.

We have carefully considered all the reviewers' comments and suggestions and have addressed each concern in our detailed responses in our revised manuscript and responses to the reviewers. Revised content is highlighted using red text.

Thank you once again for your insightful feedback and your valuable contributions to enhancing the quality of our work.

**Supporting Material:**

/attachment/892069a7dfe6af9f976a45441b7535bf572fcedb.pdf

---

### Comment · Area_Chair_snNY · 2025-03-08

Dear MIDL Reviewers, the discussion stage (March 8–14, 2025) begins today. The authors' rebuttal has been uploaded to OpenReview, and you are encouraged to engage with them for any necessary clarifications. Your participation in the discussion is greatly appreciated.

---

### Meta-Review · Area_Chair_snNY · 2025-03-15

**Recommendation:** Accept (Poster)
**Confidence:** 5

**Metareview:**

This paper attempts to address the challenge of rotation sensitivity in deep learning models applied to pathology images. To validate the methods, the rotation are added to the images for feature extraction, then K-means are used to visualize the unsupervised segmentation results.
Pros: The author provides more quantitative and qualitative evaluations, including ICC, Dice, and Cohen’s Kappa metrics for segmentation performance. The author provides point-to-point response to address reviewer concerns with additional experiments and clarifications. The proposed method achieves superior rotational consistency and segmentation quality compared with baseline methods.
Cons: The key concerns (1) technical innovation beyond the published SRENet is not justified. (2) why this method is better than geometric data augmentation to rotate and flip images during training is not justified. (3) the method achieved higher metrics value compared with baseline, but the performance is still far away from a complete rotation invariant. (4) the baseline methods used for comparison are relatively weak, which raises concerns about whether the reported improvements truly reflect a significant advancement in the field.

Thoughts: After a more thorough review of the paper, I find myself aligning more closely with Reviewers 1 and 3. Reviewer 2, who ultimately gave a strong accept, initially rated the paper as borderline due to two key concerns: (1) why CNNs were chosen over ViTs or other modern architectures, and (2) why Dice was not used as a primary evaluation metric. While these are valid points, I don't feel those are essential aspects for the paper. The Reviewer 2 changed the rating to strong acceptance after rebuttal. Reviewer 4 gave weak accept in the initial rating, but unfortunately did not participate in the rebuttal discussion, which limits a full assessment of their stance.

After considering the full paper and reviewer discussions, my primary concern aligns with Reviewers 3 and 4—specifically, whether the same improvements could be achieved simply by using data augmentation techniques like rotation and flipping. While the rebuttal attempted to justify why the proposed method is superior, I did not find the arguments fully convincing. Especially when the baseline methods are weak, and the proposed method is still far away from full rotational invariant from the performance. I agree that rotation invariance remains an important topic, particularly in pathology, making this an interesting line of research. Unfortunately, the technical innovation and methodological insights appear limited beyond the previously published SRENet, which weakens the contribution. This paper is really a boardline for me and I slightly lean to positive.